# AUTOMATIC GENERATION OF SAFETY-COMPLIANT LINEAR TEMPORAL LOGIC VIA LARGE LANGUAGE MODEL: A SELF-SUPERVISED FRAMEWORK

## ABSTRACT

Converting high-level natural-language task descriptions into formal specifications such as Linear Temporal Logic (LTL) is essential for ensuring safety in cyber-physical systems (CPS). Existing work, however, *only* optimizes translation quality without explicitly verifying the output against safety constraints. We present `AutoSafeLTL`, a self-supervised, cloud–edge–collaborative framework that automates the generation of safety-compliant LTL specifications while preserving logical consistency and semantic fidelity. A lightweight edge-side three-stage-fine-tuned LLM offers real-time conversion from natural language to LTL specifications (NL2LTL) and *guarantees* safety-critical latency and data locality. Two larger-capacity cloud-side agents then iteratively refine the alignment: 1) LLM-as-an-Aligner matches atomic propositions to safety constraints, and 2) LLM-as-a-Critic interprets counterexamples from Inclusion Check to guide corrective regeneration. This collaborative architecture provides a safety-guaranteed alignment mechanism between high-level user intent and formally verifiable system behavior, demonstrating the potential of our framework to advance AI Alignment in safety-critical domains. Our approach achieves 0% violation rates on multiple benchmarks, enabling trustworthy specification generation and verification for both AI and critical CPS applications.

## 1 INTRODUCTION

Ensuring the safety of cyber-physical systems (CPS) is central to modern system design, especially in safety-critical domains. Formal specifications like Linear Temporal Logic (LTL) Pnueli (1977) provide mathematical rigor for verification and controller synthesis Yin et al. (2024), ensuring CPS safety. The challenge lies in translating high-level, often unstructured requirements into formal specifications. However, a critical issue frequently neglected by these methods is whether the generated LTL formula inherently conflict with the safety constraints imposed on the system. We refer to this aspect as the *compliance of LTL specifications* with respect to safety constraints, and to those LTL specifications with no such conflict as *safety-compliant LTL*. Specifically, being safety-compliant means that the temporal behavior expressed by the LTL formula entirely aligns with the intended temporal behavior mandated by safety constraints. Accordingly, we focus on how to generate *safety-compliant LTL* specifications with formal guarantees to prevent critical risks to CPS.

Linear Temporal Logic (LTL) is widely used in CPS for formal verification, program analysis Pnueli (1977), and autonomous driving Sun et al. (2014). It is used to formally specify the temporal behavior in autonomous and reactive systems Hadjiloizou et al. (2024) Rodríguez & Sánchez (2024). Its adoption is growing in domains such as task planning Zhong et al. (2023b;a), reinforcement learning Li et al. (2017); Li & Belta (2019), and multi-agent systems Yang et al. (2024); Mickelin et al. (2014), due to its structured and verifiable nature. However, formulating LTL specifications is still time-consuming and error-prone, driving research on automatic translation. Recent advances improve efficiency and scalability: decomposition-based methods like SYNTHTL split complex tasks Daniel Mendoza (2024), while learning-driven approaches minimize manual effort through techniques such as Dynamic Prompt Generation Xu et al. (2024) and GraFT English et al. (2025). Beyond efficiency, research also prioritizes semantic consistency using methods like synthesis-based inconsistency detection Yan et al. (2015), human-in-the-loop interaction Cosler et al. (2023); Lahiri et al. (2023), and data augmentation for robustness Chen et al. (2023).

However, few existing methods consider whether the generated formula satisfies *external safety constraints* from the environment or system logic. Traditional approaches Nelken & Francez (1996); Hahn et al. (2023); Fuggitti & Chakraborti (2023); Liu et al. (2022) typically detect safety violations only post-deployment or during late-stage testing. In contrast, we advocate integrating formal verification into specification generation, endowing the framework with an *intrinsic safety capability*.

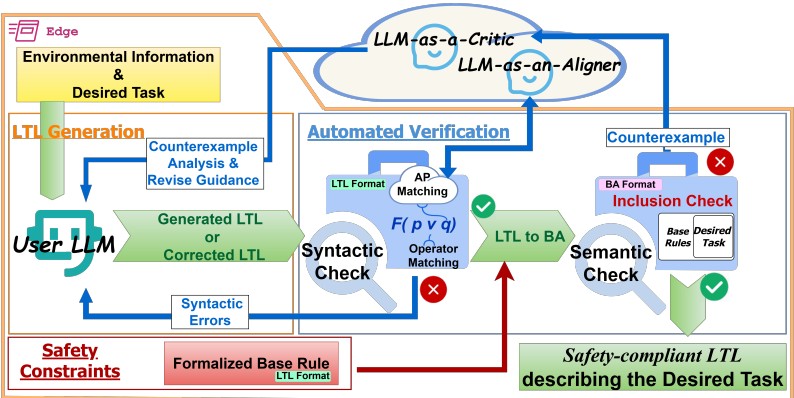

Figure 1: Overview of `AutoSafeLTL` Framework.

In this work, we present `AutoSafeLTL` (as shown in Fig. 1), a self-supervised framework that integrates automata-based verification with a cloud–edge collaborative LLM architecture to generate *safety-compliant* LTL specifications in a fully-automated manner. A *User LLM* deployed on the edge ensures minimal latency and data locality. This model is fine-tuned in three progressive stages: NL2LTL translation (S1), syntactic correction (S2), and semantic correction (S3), so that a single on-device model handles both generation and local repair. Language Inclusion checks (referred to as *Inclusion Check* in the remainder of the paper) detect early-stage violations; counterexamples and atomic-proposition mismatches are interpreted by two cloud-side agents, *LLM-as-an-Aligner* and *LLM-as-a-Critic*, which are invoked when deeper reasoning is required. This division keeps raw data on device and still leverages cloud-scale inference for rigorous semantic verification—achieving AI alignment between user intent and formally guaranteed system behavior.

**Contributions:**

- We propose a *fully automated*, end-to-end framework for NL-driven LTL specification generation, achieving, for the first time, formal safety compliance guarantees (0% violation against safety constraints) over the generated LTL. In particular, we improve existing formal verification tools by proposing an efficient algorithm for counterexample path extraction. Such an improvement enables automated analysis of counterexamples to be fed back for iterative modification of the generated LTLs, thus making the generation framework fully automated.

- We design a novel three-stage fine-tuning strategy over small language model for NL-driven LTL specification generation task. By adopting such a strategy, an 8B edge model achieves higher performance than GPT-4 in generating accurate and safety compliant LTLs, which enables the implementation of our proposed framework in resources-restricted hardware, such as edge devices.

- We establish a cloud–edge collaborative alignment framework that enforces data locality: the fine-tuned, edge-side model performs real-time generation and correction entirely on-device, ensuring that sensitive data never leaves the local environment. Meanwhile, cloud-side agents handle abstracted tasks such as AP alignment and counterexample interpretation. Such a cloud–edge framework preserves privacy and provides a principled, scalable blueprint for deploying safety-compliant NL2LTL systems in safety-critical domains.

- We construct fine-tuning and evaluation datasets to address the lack of suitable datasets for evaluating safety compliance in NL2LTL tasks. Specifically, our datasets comprise over 35k synthetic LTLs for pre-training, 200 carefully curated human–LTL pairs for fine-tuning to enhance the model's correction ability, and 300 evaluation cases, featuring navigation-style instructions with human-centered APs and rich temporal structures, enabling safety-realistic evaluation.

Extensive experiments demonstrate that `AutoSafeLTL` provides a scalable and robust solution for bridging the gap between informal requirements and safety-compliant formal specifications in CPS.

## 2 PRELIMINARY

This section introduces the background necessary for our framework, including Linear Temporal Logic (LTL), Büchi automata (BA), and Language Inclusion Checking.

### 2.1 LINEAR TEMPORAL LOGIC (LTL)

LTL serves as the foundation for many practical specification languages Pnueli (1977). The basic components of an LTL formula include Atomic Propositions (AP) and Operators (boolean and temporal). In our work, we adopt the syntax and semantics of LTL as defined in Baier & Katoen (2008); Gastin & Oddoux (2001) and recalled them here.

**Definition 1** (LTL). **(Syntax)** Let $AP$ be a finite set of atomic propositions. The set of LTL formulas over $AP$ is defined by

$$\varphi ::= \text{true} \mid a \mid \neg\varphi \mid \varphi_1 \wedge \varphi_2 \mid X\varphi \mid \varphi_1 \, U \, \varphi_2, \tag{1}$$

where $a \in AP$, and $X$, $U$ denote the *next* and *until* operators.

**(Semantics)** The semantics of an LTL formula $\varphi$ are defined over infinite words $\sigma \in (2^{AP})^\omega$, with

$$Words(\varphi) = \{\sigma \in (2^{AP})^\omega \mid \sigma \models \varphi\} \tag{2}$$

denoting the set of words satisfying $\varphi$. The full satisfaction relation is standard Baier & Katoen (2008); Gastin & Oddoux (2001) and provided in Definition 4 in the Appendix.

### 2.2 BÜCHI AUTOMATON (BA)

To bridge logical specifications with algorithmic verification, LTL formulas are typically converted into Büchi Automata (BA) Grunske (2008).

**Definition 2** (BA). A Büchi automaton (BA) is defined as a tuple $\mathcal{A} = (\Sigma, Q, \mathcal{I}, \mathcal{F}, \delta)$, where $\Sigma$ is a finite alphabet, $Q$ is a finite set of states, $\mathcal{I} \subseteq Q$ is the set of initial states, $\mathcal{F} \subseteq Q$ is the set of accepting states and $\delta : Q \times \Sigma \times 2^Q$ is the transition relation. A run of $\mathcal{A}$ on an infinite word $w = \sigma_1\sigma_2\cdots \in \Sigma^\omega$, with symbol $\sigma_i \in \Sigma$, $i \in \mathbb{N}$, starting from a state $q_0 \in Q$, is an infinite sequence of states $q_0 q_1 q_2 \ldots$ such that $(q_{j-1}, \sigma_j, q_j) \in \delta$ for all $j > 0$. The run is accepted if $q_i \in F$ for infinitely many $i$. The accepted language $L(\mathcal{A})$ consists of all infinite words over $\Sigma^\omega$ for which there exists an accepted run.

To support the subsequent Section 3.3, we introduce the notion of a *path* in a Büchi automaton (cf. Definition 5 in Appendix), which describes how an input word drives a sequence of state transitions. Since the same symbol $\sigma_i \in \sigma$ can trigger different transitions depending on the current state of the automata, there exists a set of paths over the same input word $\sigma$.

For every LTL formula $\varphi$, there exists a corresponding Büchi automaton $\mathcal{A}$ such that: $L(\mathcal{A}) = Words(\varphi)$. The aim of this conversion lies in using BA to model a system's dynamic behavior, thus enabling automata-based reasoning and verification.

### 2.3 AUTOMATA-BASED LANGUAGE INCLUSION CHECKING AMONG LTLs

The results in Esterle et al. (2020) show that safety constraints could also be interpreted as LTL (therefore BA). Therefore, the compliance of LTL specifications can be converted to an Automata-based Inclusion Check Abdulla et al. (2011).

**Definition 3** (Language Inclusion). Let $\mathcal{A}_1$ and $\mathcal{A}_2$ be two Büchi automata over the same input alphabet $\Sigma$. If every infinite word accepted by $\mathcal{A}_1$ is also accepted by $\mathcal{A}_2$, then the language of $\mathcal{A}_1$ is *included* in the language of $\mathcal{A}_2$, denoted as $L(\mathcal{A}_1) \subseteq L(\mathcal{A}_2)$:

$$L(\mathcal{A}_1) \subseteq L(\mathcal{A}_2) \iff \forall w \in \Sigma^\omega, \, w \in L(\mathcal{A}_1) \Rightarrow w \in L(\mathcal{A}_2) \tag{3}$$

This foundation enables rigorous verification of LTL specification compliance with safety constraints.

### 2.4 PROBLEM FORMULATION

Let $\mathcal{D}$ be an unstructured natural language description of a desired task, and let $\Psi = \{\psi_1, \psi_2, \ldots, \psi_n\}$ be a set of predefined safety constraints, each expressed as an LTL formula. Design a framework that synthesizes an LTL formula $\varphi$ based on $\mathcal{D}$, such that: $L(\mathcal{A}_\varphi) \subseteq \bigcap_{i=1}^n L(\mathcal{A}_{\psi_i})$, where $\mathcal{A}_\varphi$ and $\mathcal{A}_{\psi_i}$ are the Büchi automata corresponding to $\varphi$ and $\psi_i$, respectively. The objective is to generate an LTL specification $\varphi$ accroding to $\mathcal{D}$ which are guaranteed to satisfy all safety constraints in $\Psi$.

## 3 FRAMEWORK

### 3.1 OVERVIEW

To solve the problem in Section 2.2, we propose `AutoSafeLTL`, as depicted in Fig. 1, comprising two collaborative modules: *LTL Generation* and *Automated Verification*. Given *Desired Tasks* and *Environmental Information* in natural language, a lightweight edge-side *User LLM*—fine-tuned in three stages —generates and locally revises candidate LTL specifications.

The Automated Verification process consists of two stages: *Syntactic Check* and *Semantic Check*. Syntactic Check ensures grammatical validity (Definition 1). After passing the Syntactic Check, the Semantic Check ensures that the properties of state paths expressed by the candidate LTL adhere to safety constraints. In particular, both the Desired Task in LTL-format and the formalized *safety constraints* are converted into BA format for the Inclusion Check.

The principles of feedback and regeneration further ensure safety compliance. Two cloud-side *Agent LLMs* iteratively refine the LTL specifications: 1) *LLM-as-an-Aligner* ensures atomic proposition alignment; and 2) *LLM-as-a-Critic* guides corrective regeneration using Language Inclusion counterexamples. This cloud-edge synergy balances data locality and rigorous safety verification, highlighting a practical path towards AI alignment for safety-critical CPS applications. For a better illustration of the framework, we used the following running example throughout the entire session.

**Running example** Consider a traffic scenario. Table 1 presents the natural language navigation task along with a related safety constraint. Our goal is to automatically generate an LTL specification that represents the desired task while adhering to the safety constraint. This specification can then support downstream applications such as path planning and controller synthesis.

Table 1: Desired Task, Environmental Information, and Safety Constraint in a traffic scenario for the running example.

| |
| --- |
| **Desired Task:** |
| Start from the current lane, go straight for 200 meters. Then, turn right onto Maple Street and proceed for 500 meters. Finally, turn left onto Oak Street, and you will arrive at the destination after 300 meters. |
| **Environmental Information:** |
| – *Current Location:* Elm Street, Toronto; Straight lane; Speed Limit: 50 km/h; Environmental Conditions: There is a car overtaking in the right lane. |
| – *Target Location:* Oak Street, Toronto; Distance: 1 km; Estimated Arrival Time: 10 minutes. |
| – *Additional Details:* Nearby Landmarks: Springfield Museum; Traffic Signals: functioning properly. |
| **Safety Constraint in LTL Format:** |
| $G\big((\text{straight\_500m} \vee \text{right\_turn}) \rightarrow (\text{right\_turn} \rightarrow \text{straight\_500m} \rightarrow \text{left\_turn})\big) \rightarrow$ $G(\text{straight\_1km} \wedge \text{arrive\_destination})$ |

*Remark 1.* While the running example and the evaluation part in Section 5 focus on a traffic scenario[*] for illustration, our framework allows for flexible addition and adaptation of desired tasks and specifications as needed, offering high transferability and scalability across different scenarios.

*Remark 2.* While Environmental Information is not required to generate the initial LTL formula, it is essential for LLMs to *regenerate* safety-compliant specifications when violations occur (cf. Section 3.3.3), especially when the original task description is inherently unsafe. Besides, safety constraints in the form of LTL formulas are predefined, which is reasonable and supported by recent studies in traffic scenarios (e.g., Gressenbuch & Althoff (2021); Esterle et al. (2020)). However, systematically defining such constraints across diverse domains remains an important direction for future work.

### 3.2 LTL GENERATION

The LTL Generation module performs the initial NL2LTL translation from task descriptions and auxiliary environmental information on the edge. We deploy a lightweight *User LLM* that has been sequentially fine-tuned in three stages—NL2LTL translation (S1), syntactic correction (S2), and semantic correction (S3)—so that a single on-device model jointly handles generation and subsequent local repair, satisfying the CPS-driven requirement: data locality for privacy, or regulation-sensitive deployments.

Given an NL input $\mathcal{D}$ and context $E$, *User LLM* extracts an initial LTL $\varphi^{(0)}$. This output is then passed to the Automated Verification module. Fine-tuning details are illustrated in Section 4.

### 3.3 AUTOMATED VERIFICATION

Verification ensures that the generated LTL specification satisfies safety constraints. Our Automated Verification scheme, the core of the framework, comprises a *Syntactic Check* and a *Semantic Check*.

### 3.3.1 SYNTACTIC CHECK

The syntactic check consists of two steps: Atomic Proposition (AP) Matching and Operator Matching. For AP Matching, APs are extracted from safety constraints to build a library. The cloud-side *LLM-as-an-Aligner* aligns APs in the Desired Task with entries in this library and replaces them with the closest matches to ensure AP consistency. For Operator Matching, a token-based algorithm (cf. Fig. 3 in the Appendix) checks operator usage and parenthesis balance. Any detected error (along with its type and position) is fed to *User LLM* for immediate on-device correction. Prompts for the two steps are provided in the Appendix.

Now, we revisit the running example to demonstrate the syntactic check.

**Example (continued)** Having the safety constraint in LTL-format, we extract the AP library as [`right_turn`, `straight_500m`,`left_turn`,`straight_1km`, `arrive_destination`]. Then, *LLM-as-a-Aligner* is invoked to compare the LTL-formatted Desired Tasks with the AP Library. Accordingly, `go_straight_500m` in the original LTL formula is replaced with `straight_500m` from the AP Library due to their similarity, and `go_straight_200m` is replaced with the format-similar AP `straight_200m`. All other parts in the LTL formula remain unchanged, which results in a new LTL formula as follows:

$$F(\text{straight\_200m}) \wedge X\Big(G(\text{right\_turn\_Maple\_St} \rightarrow F(\text{straight\_500m} \wedge X(\text{left\_turn\_Oak\_St} \wedge$$

$$F(\text{straight\_300m}))))\Big)$$

In the Operator Matching step, both are correctly matched. Therefore, the LTL formula above is ready for the Inclusion Check in the next step. ∗

### 3.3.2 INCLUSION CHECK

After the syntactic check, the LTL formulas for the Desired Task and safety constraints are converted into BA format `Input_BA` and `Comparison_BA` for the Inclusion Check. We adopt `Spot` Duret-Lutz et al. (2022) for LTL2BA translation and `RABIT` Clemente & Mayr (2017) for Inclusion Check. Rather than simply integrating these formal tools, we adapt and enhance them to meet specific application objectives, while keeping the pipeline compatible with alternative solutions.

*Remark 3.* In practice, multiple safety constraints may exist as separate LTL formulas. Instead of combining them, our framework verifies each formula in parallel, supports adding new rules dynamically, and allows prioritizing checks by importance to provide different levels of safety assurance. To provide more intuition for the inclusion check, we now revisit the running example.

**Example (continued)** Both the Safety Constraint and Desired Task are converted into Büchi automata (see Fig 4 in Appendix) and then subjected to a Ramsey-based inclusion check. The process terminates once a counterexample is found, which in this case is !`straight_200m`, as summarized in Table 2. This counterexample is then used to guide LTL correction in the next step.

Table 2: Counterexample found in the first round of Inclusion Check for the running example.

**Counterexample:**
Aut A (after light preprocessing): of Trans. 47, of States 11.
Aut B (after light preprocessing): of Trans. 8, of States 3.
Counterexample: !straight_200m
Not included.
Time used(ms): 38.
∗

### 3.3.3 COUNTER-EXAMPLE-GUIDED AUTOMATIC CORRECTION

In Language Inclusion Check, a single counterexample word $\sigma$ can be found if there exist any violations, but $\sigma$ may correspond to multiple dynamic behaviors, as the same symbol $\sigma_i \in \sigma$ can trigger different transitions depending on the current state of the automata. Consequently, identifying a concrete counterexample path - a sequence of state transitions combined with the driving symbol - is essential for accurately locating the source of the violation. This path serves as a precise semantic witness for non-inclusion and provides actionable guidance for correcting the LTL specification. To this end, we formally define the notion of a counterexample path and prove its existence under non-inclusion; the proof is provided in the Appendix (Theorem 1).

Based on Theorem 1, we integrate path extraction into the Inclusion Checking process via the Comparison Path Storage Algorithm 1. The worst case complexity of extracting a counterexample path is $O(n_1 \times n_2 \times m)$, where $n_1$ and $n_2$ are the state sizes of the two Büchi automata and $m$ is the alphabet size. The formal proposition 1 and its proof A.2 are provided in the Appendix, demonstrating the usability of this theoretical foundation, which constitutes a key contribution of our work.

---

**Algorithm 1** Comparison Path Storage

---

**Input**: Two automata $\mathcal{A}_1$, $\mathcal{A}_2$, maximum search depth $maxdepth$
**Output**: A counterexample path string, or `null` if no counterexample is found

1: Initialize $post[state][symbol]$ for state transitions
2: Initialize queue with initial state pairs $(state1, state2)$ from $\mathcal{A}_1$, $\mathcal{A}_2$
3: Initialize visited set to track explored state pairs
4: **while** queue is not empty **do**
5:     Dequeue $(state1, state2)$
6:     **if** $state1$ is in visited **then**
7:       **if** $state2$ is in visited$[state1]$ **then**
8:         **continue**
9:       **end if**
10:     **end if**
11:     Add $state2$ to visited$[state1]$
12:     **for** each symbol $s$ **do**
13:       $nextStates1 \leftarrow post[state1][s]$
14:       $nextStates2 \leftarrow post[state2][s]$
15:       **if** $nextStates1 \neq nextStates2$ **then**
16:         Build and store the path from $(state1, state2)$
17:         **return** the path
18:       **end if**
19:     **end for**
20: **end while**

---

To maximize the utility of counterexample paths, we introduce *LLM-as-a-Critic*. We discovered that enabling one LLM to understand another LLM can overcome the LLM's lack of knowledge in formal verification and format conversion. First, we let *LLM-as-a-Critic* directly analyze the counterexample path and provide suggestions for modifying the input LTL. Then, we pass the output of *LLM-as-a-Critic* to *User LLM*, which has access to Environmental Information, to perform the actual LTL modifications. Detailed prompts used for invoking LLMs here are presented in the Appendix. This process not only enhances the extraction of counterexample information but also allows for more effective modifications by incorporating Environmental Information (cf. Section 5.3 for the ablation experiment result).

To illustrate this process, we now revisit the running example.

**Example (continued)** First, we input the counterexample path into the *LLM-as-a-Critic*. The obtained violation analysis and correct guidance are illustrated in Table 14 of the Appendix and then be fed to the User LLM to modify the LTL. Accordingly, we obtain:

$$G\big((\text{starting\_location}) \rightarrow F(\text{straight\_200m})\big) \wedge X\Big(G\big(\text{right\_turn\_Maple\_St} \rightarrow F\big(\text{straight\_500m} \wedge$$
$$X\big(\text{left\_turn\_Oak\_St} \wedge F(\text{straight\_300m})\big)\big)\big)\Big)$$

Although the LTL above still fails the inclusion check, after iterating the automated verification loop three times, the final LTL that adheres to the rules is obtained as follows:

$$X(\text{destinationLeftOnMapleStreet}) \wedge G\big((\text{location\_start}) \rightarrow \big(F(\text{straight\_200m}) \vee G(\neg\text{straight\_200m})\big)\big) \wedge$$
$$X\Big(G\big(\text{right\_turn\_Maple\_St} \rightarrow F\big(\text{straight\_500m} \wedge X\big(\text{left\_turn\_Oak\_St} \wedge F(\text{straight\_300m})\big)\big)\big)\Big)$$

Our framework prioritizes safety compliance over strict semantic fidelity to the Desired Task: when the original route violates safety, it is reordered with a safe alternative that still reaches the same destination. In this example, the initial LTL formula already satisfies the latter part of the Safety Constraint, so the segment from "turn right onto Maple Street" remains unchanged. And because the Safety Constraint requires the journey to begin by proceeding straight or making a right turn, we add the `straight_200m` restriction. It also stipulates that the destination is reached after a left turn followed by straight driving. Using Environmental Information, we identify that the last left turn occurs after entering Maple Street and therefore add $X(\text{destinationLeftOnMapleStreet})$ to ensure correct arrival.           *

*Remark 4.* Our ideal objective is to search for an LTL specification that satisfies all safety constraints while making the minimal semantic modification to the Desired Task. In practice, we set a maximum number of repair rounds and a timeout to prevent infinite correction loop for the Desired Task over which a safety-compliant case can't be found.

Once the Desired Task in LTL format passes the semantic check, a safety-compliant LTL $\varphi^{(k)}$ after $k$ rounds with respect to predefined safety constraint is obtained. Note that based on our framework's mechanism and the Inclusion Check's mathematical rigor, it is guaranteed that *any LTL that is not safety-compliant will not be output*, which is critical for its involvement in subsequent applications. This would also be shown by the experiment in Section 5.3.

## 4 FINETUNING APPROACH

### 4.1 CONSTRUCTION OF DATASETS

A major limitation of existing NL2LTL datasets is their abstract APs (e.g., SYNTCOMP Jacobs et al. (2024)), which lack practical context. We emphasize human-centered APs to aid counterexample-driven correction. Since our goal is to fine-tune a lightweight LLM, the dataset must be compact yet effective.

We chose the traffic navigation as the primary scenario. To support the three-stage-fine-tuning process of our framework, we construct dedicated datasets for each stage, following a generate–filter–refine paradigm that ensures syntactic correctness, semantic alignment, and safety compliance, respectively.

1) For stage-1, We designed 50 navigation instruction templates and expanded them into NL–LTL pairs using GPT-4. Generated formulas were validated with SPOT Duret-Lutz et al. (2022), retaining only syntactically correct ones. We first perform a manual semantic check to ensure each natural language instruction accurately reflects the intent of its corresponding LTL formula. Next, Then, we standardize and refine the wording of all instructions using a simplified prompt template, making their style consistent with downstream applications. The final result is a set of 200 high-quality Instruction-LTL training pairs. (see Table 8 in the Appendix)

2) For stage-2, erroneous LTLs from generation were annotated with error types and paired with corrected versions passing SPOT, forming 225 Instruction–CorrectedLTL pairs across 69 error types.

3) For stage-3, using RABIT Clemente & Mayr (2017), we collected LTLs that initially failed Inclusion Check but passed after manual repair. Each case produced an Instruction–Counterexample–CorrectedLTL triple, totaling 50 examples with concrete counterexamples. These rare and time-consuming cases make the collection especially valuable for training and evaluation; we demonstrate its effectiveness in Section 4.3.

For evaluation, we construct a set of 300 navigation tasks that are both naturalistic and structurally diverse, featuring previously unseen combinations of atomic propositions. This dataset is completely held out from all training stages and serves as the evaluation benchmark for all LTL specification generation tasks in our work. A subset of this benchmark is shown in the Technical Appendix.

### 4.2 MODEL FINE-TUNING

We adopt LLaMA-3-8B-Instruct as the edge-side *User LLM*. To maintain on-device efficiency, we employ QLoRA with LoRA adapters, which update only low-rank weights at each stage. This enables incremental injection of repair knowledge while avoiding catastrophic drift. Implementation details, including adapter configuration, optimization hyperparameters, and training setup, are provided in Table 9 in the Appendix.

### 4.3 PROOF OF EFFICIENCY

To validate the effectiveness of our progressive fine-tuning strategy, we conduct a series of ablation experiments to assess the impact of each stage. Stage-1 focuses on improving the accuracy (i.e., the correctness of initial NL-to-LTL translation before any repair or refinement). Stage-2 enhances the model's ability to generate syntactically correct formulas and efficiently self-correct. Stage-3 further equips the model to align with safety constraints and enhances its ability to perform semantic repair.

**Stage-1** Similar to Chen et al. (2024), we adopt binary accuracy (i.e., 100% correct or not) to measure raw translation performance, although this is not our primary focus. The evaluation is conducted on the 300-sample evaluation benchmark with 6 different settings.

- **LLaMA-Origin**: the base LLaMA-3-8B-Instruct model without fine-tuning.
- **GPT-4**: zero-shot prompting using the same task format.
- **LLaMA-200-raw**: LLaMA fine-tuned on 200 pairs filtered only by `SPOT` for syntactic validity (no semantic cleaning).
- **LLaMA-200-clean**: LLaMA fine-tuned on 200 pairs after both syntactic validity and manual semantic refinement.
- **LLaMA-35k-lifted**: LLaMA fine-tuned on 35,000 NL–LTL pairs adapted from the 28k lifted STL corpus in Chen et al. (2024). Each STL formula is first converted into an

LTL specification, where atomic propositions are rewritten to represent traffic navigation instructions. The corresponding natural language descriptions are then generated by GPT-4.

- **LLaMA-200-refined**: LLaMA fine-tuned on 200 cleaned pairs with deeper refinement and instruction-level optimization (cf. Section 4.1).

The results (cf. Table 5) highlight the importance of data quality: simply cleaning a 200-example corpus (LLaMA-200-clean) narrows the gap to GPT-4, while a refined dataset makes LLaMa outperform GPT-4 with a lightweight 8B model. Conversely, a much larger but less targeted corpus (LLaMA-35k-lifted) degrades performance confirming that *task-specific, semantically aligned data outweighs sheer quantity* for compact LLMs in structured specification generation. Among Stage-1 configurations, LLaMA-200-refined achieves the highest accuracy and is therefore selected for subsequent fine-tuning.

**Stage-2** To evaluate the effect of syntactic correction fine-tuning, we measure the *average number of syntax correction iterations* required to produce a syntactically correct LTL formula. This metric reflects both the model's initial correctness and its ability to efficiently repair errors when necessary. As shown in Table 5, `AutoSafeLTL` achieves the best results, reducing the average to **0.56** versus 1.72 at Stage-1—a 67% drop—outperforming LLaMA-Origin and GPT-4. This shows Stage-2 fine-tuning effectively instills syntactic knowledge, with many formulas correct on the first attempt.

**Stage-3** To evaluate the effect of semantic correction fine-tuning, we measure the *average number of semantic correction iterations*—i.e., revisions required to achieve an `Included` status from a safety-violating LTL formula. This metric reflects the model's *intrinsic safety capability* and ability to efficiently repair non-compliant formulas.

The final stage further enhances semantic alignment through verification-guided correction. `AutoSafeLTL (Stage-3)` achieves the best performance on both syntax (**0.51** fixes) and semantic repair (**4.3** fixes), outperforming all baselines including GPT-4. And since Stage-3 introduces perturbations during counterexample-driven refinement, which may reduce surface-level matching but improve safety compliance, its accuracy (93.6%) is slightly below the Stage-1's peak, yet still outperforms GPT-4. This aligns with our main goal—ensuring safety-compliant LTLs.

Table 5: Stage-wise performance of our framework. Accuracy reflects the initial generated LTL's quality; Syn./Sem. Fixes indicate average iterations for syntax/semantic repair.

| Model | Accuracy (%) | Syn. Fixes | Sem. Fixes |
|---|---|---|---|
| LLaMA-Origin | 86.7 (260/300) | 2.1 | 16.2 |
| GPT-4 | 92.3 (277/300) | 0.85 | 7.8 |
| AutoSafeLTL (Stage-1) | **95.7 (287/300)** | 1.72 | 7.1 |
| LLaMA-200-raw | 91.3 (274/300) | – | – |
| LLaMA-200-clean | 92.7 (278/300) | – | – |
| LLaMA-35k-lifted | 90.0 (270/300) | – | – |
| AutoSafeLTL (Stage-2) | 94.3 (283/300) | 0.56 | 6.8 |
| **AutoSafeLTL (Stage-3)** | 93.6 (281/300) | **0.51** | **4.3** |

## 5 FRAMEWORK EVALUATION

### 5.1 SETUP

We evaluate our framework's end-to-end effectiveness under a cloud–edge collaborative architecture, using *violation counts*, i.e., safety violations detected via Inclusion Check against predefined safety constraints, as the primary metric. Lower violation counts indicate better safety compliance.

We deploy `GPT-4` as cloud-edge Agent LLMs without fine-tuning, accessed via API and prompt engineering. Our evaluation is conducted on the same evaluation benchmark (cf. Section 4.1).

We compare our approach against several baselines:

- **LLaMA&GPT-4**: zero-shot prompting using the same template as our method.
- **`nl2spec`** Cosler et al. (2023): a LTL extraction tool. We adopt its default settings—`GPT-3.5-turbo` as the backend, `minimal` extraction mode, and `Number of tries` set to 3.
- **AutoSafeLTL**: our full pipeline under the collaborative cloud–edge setting.

Notably, many real-world navigation instructions in our dataset lack a rigid logical structure, resulting in very low translation coverage by `nl2spec` (less than 5% of original inputs produce valid LTL). For fair comparison, we use `GPT-4o` to rewrite all Desired Tasks into logically enhanced forms before `nl2spec` translation. These rephrased instructions serve as its evaluation input.

Table 6: Initial violation rate comparison. **AutoSafeLTL\*** uses only *User LLM* without automated verification.

|  | **llama** | **gpt-4** | **nl2spec** | **autosafeltl\*** |
|---|---|---|---|---|
| violation (%) | 100 | 100 | 100 | **98** |

Table 7: violation rate and interaction rounds on the benchmark dataset. **B** denotes the automated verification component; **C**, the *llm-as-an-aligner*; and **D**, the *llm-as-a-critic*.

|  | LLaMA+B | GPT-4+B | nl2spec+B | AutoSafeLTL | AutoSafeLTL-C | AutoSafeLTL-D |
|---|---|---|---|---|---|---|
| Violation (%) | 0 | 0 | 0 | 0 | 0 | 0 |
| Avg. interaction | 16.2 | 7.8 | 6.5 | **4.3** | 15.0 | 11.0 |

We also conduct ablation studies to investigate the necessity of both *User LLM* and *Agent LLM* in our framework. These variants isolate the contribution of each model and allow us to assess how collaboration between cloud and edge components influences final safety outcomes.

## 5.2 QUALITY OF INITIAL TRANSLATION.

We first evaluate the *initial violation rate* on the benchmark—i.e., the proportion of initially generated LTL formulas that conflict with safety compliance. Each initial LTL formula undergoes a single Inclusion Check: those marked INCLUDED are considered compliant, all others are violations.

In previous research on LTL generation, compliance has remained an overlooked aspect. As shown in Table 6, prior approaches exhibit a **100%** violation rate, indicating that none of their generated LTL formulas initially conform to the safety constraints. In contrast, our `AutoSafeLTL*`—a variant that uses only the *User LLM* without invoking the automated verification loop—achieves a violation rate of **98%**, successfully producing safety-compliant LTL at generation time.

This result, though modest, is significant: it demonstrates that our progressive fine-tuning pipeline enables the model to internalize safety priors and generate compliant LTLs out of the box—something no baseline achieves. It also underscores the necessity of embedding safety awareness during training, especially for applications in safety-critical CPS.

## 5.3 QUALITY OF VERIFIED TRANSLATION AND ABLATION STUDY

Next, we evaluate the effectiveness of our safety assurance component *Automated Verification module*. An LTL formula is *safety-compliant* if it passes the final Inclusion Check. Additionally, we measure the modification iterations needed for compliance as a supplementary efficiency metric.

We apply our full verification pipeline (`+B`) to LTLs generated by multiple baselines, including `LLaMA`, `GPT-4`, and `nl2spec`. As shown in Table 7, all models achieve a violation rate of **0%**, confirming the generalizability of the verification process. However, our `AutoSafeLTL` achieves this with the fewest average interaction rounds (**5.7**), significantly outperforming `GPT-4+B` (7.8) and `LLaMA+B` (16.2), demonstrating higher alignment efficiency with formal safety constraints.

To further understand the role of each cloud-side agent, we perform ablation experiments by selectively removing *LLM-as-an-Aligner* (`AutoSafeLTL-C`) and *LLM-as-a-Critic* (`AutoSafeLTL-D`). Both variants still achieve full compliance, but at the cost of more iterations (15.0 and 11.0, respectively), highlighting the complementary importance of both agents in minimizing verification overhead.

Together, these results validate the effectiveness and efficiency of our collaborative verification loop. Our approach provides verified safety-compliance with formal guarantees for the generated LTLs, making it a practical and reliable solution for safety-critical applications.

## 6 CONCLUSION

We propose `AutoSafeLTL`, a self-supervised, fully-automated framework that integrates LLM-based specification generation with formal verification to ensure safety-compliance in LTL outputs. By combining progressive fine-tuning and automated verification mechanisms, our system achieves full safety compliance with zero violations. While the formal verification tools are a modular component of our framework, their occasional failures on complex inputs may impair overall performance and system reliability. Future directions include building learning-enhanced custom Inclusion Checkers for greater verification robustness and scaling the framework to support a wider range of scenarios and more diverse safety specifications beyond navigation-style tasks.

## A  ETHICS STATEMENT

This work does not involve human subjects, personally identifiable information, or sensitive user data. All datasets used in this paper were either publicly available or synthetically constructed by the authors, ensuring compliance with privacy and legal standards. The authors affirm that the research adheres to the Code of Ethics and upholds principles of fairness, integrity, and responsible AI research.

## B  REPRODUCIBILITY STATEMENT

To facilitate repeatability, we provide a detailed description of our experimental settings in this part.

### B.1  CONSTRUCTION AND ENVIRONMENT SET UP

The folder "AutoSafeLTL" contains code for constructing an automatic system to transform natural language into safe LTL. In particular, the file "AutoSafeLTL.py" includes all functions that are necessary for the transformation process. To execute this script, the following software and packages are required:

1. **Java**: Installing Java (24). The official installation guide can be found at:
   `https://docs.oracle.com/en/java/javase/24/install/installation-jdk-microsoft-windows-platforms.html` (Windows) and `https://docs.oracle.com/en/java/javase/24/install/installation-jdk-macos.html` (macOS).

2. **Python**: Installing Python (3.11.6). The official installation guide can be found at:
   `https://docs.python.org/3.11/using/windows.html` (Windows) and `https://docs.python.org/3.11/using/mac.html` (macOS).

3. **Python module**: Installing requests, torch, transformers and selenium. Refer to PyCharm's official guidelines at:
   `https://www.jetbrains.com/help/pycharm/2023.2/installing-uninstalling-and-upgrading-packages.html`.

4. **RABIT 2.5.0**: Decompressing **rabit250-new.zip** under the folder "AutoSafeLTL" to the original folder.

5. **SPOT 2.14**: Deploying the Spot tool by executing the following commands in Windows PowerShell with administrator privileges:

   - Install WSL subsystem:

   ```
   wsl --install
   ```

   - Install required dependencies:

   ```
   sudo apt update
   sudo apt install -y build-essential autoconf automake libtool pkg-config bison
        ↪ flex swig python3 python3-dev cython3 libgmp-dev libltdl-dev libboost-dev
        ↪ python3-numpy
   ```

   - Download and extract Spot source code:

   ```
   wget https://www.lrde.epita.fr/dload/spot/spot-2.14.tar.gz
   tar -xf spot-2.14.tar.gz
   cd spot-2.14
   ```

   - Compile and install Spot tool (Replace "/home/yourname/spot" with your custom installation path):

   ```
   ./configure --prefix=/home/yourname/spot
   make -j4
   make install
   ```

   - Configure environment variables:

```
echo 'export PATH=$HOME/spot/bin:$PATH' >> ~/.bashrc
echo 'export LD_LIBRARY_PATH=$HOME/spot/lib:$LD_LIBRARY_PATH' >> ~/.bashrc
source ~/.bashrc
```

- Verify installation success:

```
ltl2tgba -V
```

## B.2   MAIN FUNCTION STRUCTURE

The AutoSafeLTL_Method function is structured as follows:

```
AutoSafeLTL_Method
|-- Initialization
|   |-- Input: Comparison LTL Formula
|   |-- Output: Atomic Proposition Library
|-- Preprocessing
|   |-- Atomic Proposition Standardization (gpt_replace_AP)
|-- Main Loop
|   |
|   |-- Phase 1: Formal Conversion
|   |   |-- LTL -> BA Conversion (run_ltl2tgba)
|   |   `-- HOA -> BA Conversion (hoa_to_ba)
|   |
|   |-- Phase 2: Safety Verification
|   |   |-- RABIT Language Inclusion Check
|   |   |-- Pass -> Exit Loop
|   |   `-- Fail -> Proceed to Correction
|   |
|   `-- Phase 3: AI Correction
|       |-- GPT Problem Analysis (gpt_understand_rabit_output)
|       |-- Local Correction Generation (local_correct_ltl)
|       `-- Syntax Verification (correct_ltl_formula)
|
`-- Result Output
    |-- Verification Status: Included / Not Included
    `-- Final LTL Formula
```

## B.3   CONDUCTING PROCESS

To conduct the process, one should:

- Check RABIT and SPOT files exist
- Input your own API key in the file "AutoSafeLTL.py" as shown in Fig. 2
- Run `AutoSafeLTL.py` file

After the process is completed, one gets a JSON file "SafeLTL" that contains safety compliant LTL data.

```
136    # GPT model configuration
137    gpt_gen = LLM_Generator(
138        mode='gpt',
139        api_key="",
140        model_name="gpt-4"
141    )
```

Figure 2: Input your own API key at line 139

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

# A APPENDIX

## A.1 USE OF LARGE LANGUAGE MODELS

Large Language Models (LLMs) were used solely to refine the language of this manuscript. Specifically, LLMs were used to polish the manuscript by improving readability, ensuring clarity, and enhancing linguistic consistency. In addition, we employed LLMs in controlled settings to assist with data generation (e.g., expanding instruction templates into NL–LTL pairs) and prompt-based rewriting for dataset unification.

It is important to emphasize that the LLMs did not contribute to the conceptualization of the research problem, theoretical development, or the design of experiments. All core research ideas, methodological innovations, proofs, and analyses were independently developed by the authors. The LLM contributions were strictly limited to linguistic refinement and structured data augmentation under human supervision.

The authors take full responsibility for the content of this paper, including any text or data generated with LLM assistance. All LLM outputs were carefully reviewed, validated, and, when necessary, modified by the authors to ensure correctness and compliance with ethical guidelines.

## A.2 SUPPLEMENTARY MATERIALS

**Definition 4** (LTL). **(Syntax)** Let $AP$ be a finite set of Atomic Propositions. The set of LTL formulas over $AP$ is defined inductively by the following grammar:

$$\varphi ::= \text{true} \mid a \mid \neg\varphi \mid \varphi_1 \wedge \varphi_2 \mid X\varphi \mid \varphi_1 U \varphi_2, \tag{4}$$

where $a \in AP$ is an Atomic Proposition, temporal operators such as $X$ (next), $U$ (until) can be used for dynamical properties.

**(Semantics)** Let $\varphi$ be an LTL formula, the semantics of $\varphi$ is defined over infinite words $\sigma \in (2^{AP})^\omega$, with $Words(\varphi) = \left\{\sigma \in \left(2^{AP}\right)^\omega \mid \sigma \models \varphi\right\}$ being the LT property induced by $\varphi$. Let $\sigma = A_0 A_1 A_2 \ldots$ (symbol $A_i \in AP$), where the satisfaction relation $\models \subseteq (2^{AP})^\omega \times \text{LTL}$ is defined as follows:

- $\sigma \models \text{true}$,

- $\sigma \models a$ iff $a \in A_0$,
- $\sigma \models \neg\varphi$ iff $\sigma \not\models \varphi$,
- $\sigma \models \varphi_1 \wedge \varphi_2$ iff $\sigma \models \varphi_1$ and $\sigma \models \varphi_2$,
- $\sigma \models X\varphi$ iff $\sigma[1...] = A_1 A_2 \ldots \models \varphi$,
- $\sigma \models \varphi_1 U \varphi_2$ iff $\exists j \geq 0$ such that $\sigma[j...] \models \varphi_2$ and for all $0 \leq i < j$, $\sigma[i...] \models \varphi_1$.

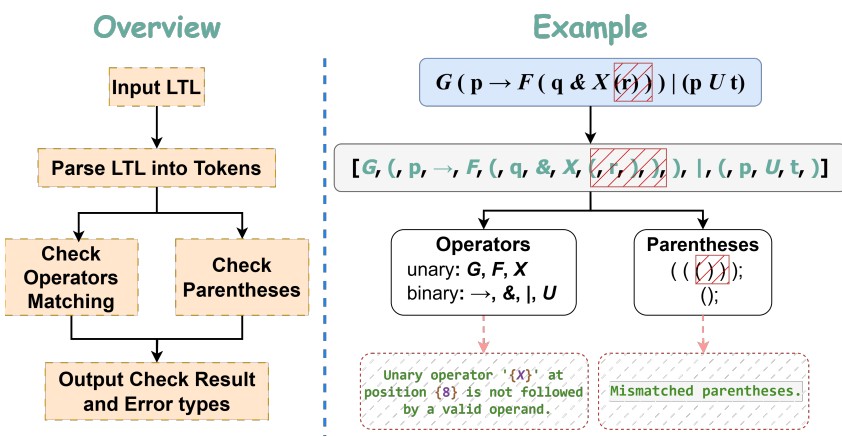

Figure 3: Overview of Operator Matching in the Syntactic Check.

**Definition 5** (Path). Let $\mathcal{A} = (\Sigma, Q, \mathcal{I}, \mathcal{F}, \delta)$ be a Büchi automaton and let *word* $\sigma = \sigma_0 \sigma_1 \sigma_2 \ldots \in \Sigma^w$ with symbol $\sigma_i \in \Sigma$, $i \in \mathbb{N}$. A *path* $\rho$ of $\mathcal{A}$ over $\sigma$ is a state-and-word sequence:

$$q_0 \xrightarrow{\sigma_0} q_1 \xrightarrow{\sigma_1} q_2 \xrightarrow{\sigma_2} \cdots$$

where $q_0 \in \mathcal{I}$ is an initial state and for each $i \geq 0$, $(q_i, \sigma_i, q_{i+1}) \in \delta$. A path $\rho$ is *simulable* w.r.t. $\sigma$ if: 1) it traverses every $\sigma_i \in \sigma$ and the corresponding transitions $(q_i, \sigma_i, q_{i+1}) \in \delta$; 2) it ends in or visits some accepting state $q_f \in \mathcal{F}$ infinitely often. The set of paths over $\sigma$ in $\mathcal{A}$ is denoted by $\mathcal{P}_\sigma$, where:

$$\mathcal{P}_\sigma = \{q_0 \xrightarrow{\sigma_0} q_1 \xrightarrow{\sigma_1} \cdots \mid q_i \in Q,\ \sigma_i \in \sigma,\ (q_i, \sigma_i, q_{i+1}) \in \delta \tag{5}$$

**Theorem 1** (Existence of Counterexample Path). Let $\mathcal{A}_1 = (\Sigma, Q_1, \mathcal{I}_1, \mathcal{F}_1, \delta_1)$ and $\mathcal{A}_2 = (\Sigma, Q_2, \mathcal{I}_2, \mathcal{F}_2, \delta_2)$ be two Büchi automata. If there exists an infinite counterexample word $\sigma = \sigma_1(\sigma_2)^\omega \in \Sigma^\omega$, where $\sigma_1 \in \Sigma^*$ is a finite prefix, and $\sigma_2 \in \Sigma^+$ is a non-empty finite loop, such that $\sigma$ is accepted by $\mathcal{A}_1$ but rejected by $\mathcal{A}_2$ (i.e., witnessing $L(\mathcal{A}_1) \not\subseteq L(\mathcal{A}_2)$), then a finite counterexample path $\rho \in \mathcal{P}_\sigma$ on $\sigma$ exists, with $\mathcal{P}_\sigma$ as in equation 5, which is simulable in $\mathcal{A}_1$ but not simulable in $\mathcal{A}_2$.

*Proof.* Let $\sigma = \sigma_1(\sigma_2)^\omega \in L(\mathcal{A}_1) \setminus L(\mathcal{A}_2)$ be a counterexample word. By definition, there exists an accepting path of $\mathcal{A}_1$ on $\sigma$:

$$\rho_1 : q_{1,0} \xrightarrow{\sigma_1} q_{1,c} \xrightarrow{\sigma_2} q_{1,c} \xrightarrow{\sigma_2} \cdots,$$

where $\sigma_2$ visits accepting state(s) infinitely. Assuming that there is no finite counterexample path $\rho$ on word $\sigma$ that is simulable in $\mathcal{A}_1$ but not simulable in $\mathcal{A}_2$ exists. Then, for all $n \in \mathbb{N}$, the finite path $\rho^{(n)} = \sigma_1(\sigma_2)^n$ must be simulable by $\mathcal{A}_2$, as in Definition 2.4; that is, $\mathcal{A}_2$ can simulate a path:

$$\rho_2 : q_{2,0} \xrightarrow{\sigma_1} q_{2,c}^{(n)} \xrightarrow{\sigma_2^n} \tilde{q}_{2,c}^{(n)}, \text{with } q_{2,c}^{(n)}, \tilde{q}_{2,c}^{(n)} \in Q_2,$$

Since $\sigma \notin L(\mathcal{A}_2)$, every infinite path of $\mathcal{A}_2$ on $\sigma$ either: 1) Fails to being driven by $\sigma_1$ (terminating at some $i \leq |\sigma_1|$), or 2) Processes $\sigma_1(\sigma_2)^\omega$ but fails to satisfy the Büchi condition (an $\omega$-regular acceptance condition) (Büchi,1996).

Case 1 (Finite Rejection): Suppose $\mathcal{A}_2$ cannot be driven by $\sigma_1$ fully. Let $i \leq |\sigma_1|$ be the minimal position where no transition in $\mathcal{A}_2$ matches $\mathcal{A}_1$'s step $q_{1,i} \xrightarrow{\sigma_{i+1}} q_{1,i+1}$. The finite path $q_{1,0} \xrightarrow{\sigma_1} q_{1,1} \xrightarrow{\sigma_2} \cdots \xrightarrow{\sigma_i} q_{1,i}$ can be found in $\mathcal{A}_1$, but $\mathcal{A}_2$ lacks a corresponding transition at step $i$. This contradicts the assumption that all finite paths $\rho^{(n)}$ are simulable in $\mathcal{A}_2$.

Case 2 (Infinite Rejection): If $\mathcal{A}_2$ can be driven by $\sigma_1$ fully, let $q_{2,c}$ denote its state after $\sigma_1$. The infinite suffix $\sigma_2^\omega$ induces a path in $\mathcal{A}_2$:

$$\rho_2^\omega : q_{2,c} \xrightarrow{\sigma_2} q_{2,c}^{(1)} \xrightarrow{\sigma_2} q_{2,c}^{(2)} \xrightarrow{\sigma_2} \cdots ,$$

forming an infinite sequence of states. As $\sigma \notin L(\mathcal{A}_2)$, $\rho_2^\omega$ must fail the Büchi condition. By the pigeonhole principle ( Jukna, 2011), there exists a strongly connected component (SCC) $C \subseteq Q_2$ visited infinitely often in $\rho_2^\omega$, containing no accepting states. Let $k \in \mathbb{N}$ be the minimal number such that $\mathcal{A}_2$'s path on $\sigma_2^k$ closes a cycle within $C$. Then, the finite path $q_{1,0} \xrightarrow{\sigma_1} q_{1,c} \xrightarrow{\sigma_2} \cdots \xrightarrow{\sigma_2} q_{1,c}$ ($k+1$ states) can be found in $\mathcal{A}_1$, but $\mathcal{A}_2$'s corresponding path $\rho_2^w$ either terminates prematurely or enters $C$. Both violate the assumption that $\mathcal{A}_2$ simulates all finite paths.

In either case, the existence of $\rho$ which is simulable in $\mathcal{A}_1$ but not simulable in $\mathcal{A}_2$ is unavoidable, contradicting the assumption. □

**Definition 6** (Supergraph in Ramsey-Based Inclusion Checking). Let $\mathcal{A}_1 = (\Sigma, Q_1, \mathcal{I}_1, \mathcal{F}_1, \delta_1)$ and $\mathcal{A}_2 = (\Sigma, Q_2, \mathcal{I}_2, \mathcal{F}_2, \delta_2)$ be two Büchi Automata over the same alphabet $\Sigma$. A *supergraph* $\mathcal{G}(V, E)$ is a finite directed graph constructed to capture the combined behaviors of $\mathcal{A}_1$ and $\mathcal{A}_2$. The set of nodes $V \subseteq Q_1 \times Q_2$ consists of all pairs of states $(q_1, q_2)$ where $q_1 \in Q_1$ and $q_2 \in Q_2$, the set of edges $E \subseteq V \times \Sigma \times V$ captures synchronized transitions under the same input symbol $a \in \Sigma$, s.t.: If $(q_1, a, q_1') \in \delta_1$ and $(q_2, a, q_2') \in \delta_2$, then $((q_1, q_2), a, (q_1', q_2')) \in E$.

**Proposition 1.** Let $\mathcal{A}_1 = (\Sigma, Q_1, \mathcal{I}_1, \mathcal{F}_1, \delta_1)$ and $\mathcal{A}_2 = (\Sigma, Q_2, \mathcal{I}_2, \mathcal{F}_2, \delta_2)$ be two Büchi automata. If a counterexample word $\sigma \in \Sigma^\omega$ exists (as defined in Theorem 1), then the counterexample path $\rho$ (as described in Theorem 1) can be identified in time $O(n_1 \times n_2 \times m)$ in the worst-case, where $n_1 = |Q_1|$, $n_2 = |Q_2|$ and $m = |\Sigma|$, where $|\cdot|$ being the cardinality of a set.

*Proof.* Define $n_1 = |Q_1|$, $n_2 = |Q_2|$, and let $m = |\Sigma|$. The supergraph $\mathcal{G}(V, E)$ as defined in Definition 6 is constructed over state pairs $(q_1, q_2) \in Q_1 \times Q_2$, with: $|V| \leq n_1 \times n_2$. We use the Breadth-first search(BFS) Algorithm to traverse from all state pairs at most once to detect a counterexample. Assume the size of the alphabet is fixed and finite, i.e., $|\Sigma| = m$, then each pair requires exactly $m$ comparisons. Therefore, the time complexity of transition checks $O(\cdot)$ is bounded by: $O(\cdot) \leq O(|V| \times m) \leq O(n_1 \cdot n_2 \cdot m)$.

When a mismatch is found, the counterexample path is constructed via backtracking from the visited state map (as also illustrated in Algorithm 1). The length of this path is at most the depth of the BFS tree, i.e., $n_1 \times n_2$, and can be performed in linear time. Since path construction occurs once and after all comparisons, it does not affect the asymptotic bound.

Even if all vertices must be validated for acceptance conditions, the total operations remain dominated by $|V|$. Thus, the counterexample trace is obtained in time: $O(n_1 \times n_2 \times m)$. □

Table 8: Statistics of stage-1 dataset

| Metric | Value range (average) |
|---|---|
| Number of atomic propositions (APs) | 4–8 (avg. 6.35) |
| Number of temporal operators | 8–20 (avg. 13) |
| Number of nested temporal operators | 2–13 (avg. 8) |
| Nesting depth | 4–9 (avg. 5.45) |

## A.3 EXPERIMENTAL PROMPTS

Tables below show the detailed prompts used for invoking LLMs, including *User LLM*, *LLM-as-a-Critic* and *LLM-as-an-Aligner*, as mentioned in Section 3.

Table 9: Hyperparameters and training configuration for model fine-tuning

| | |
|---|---|
| Base model | LLaMA-3-8B-Instruct |
| Quantization | QLoRA (4-bit) |
| LoRA target modules | k_proj, q_proj, v_proj, o_proj |
| Rank $r$ / scaling $\alpha$ | 32 / 32 (gain=1) |
| Epochs / batch size | 5 / 16 |
| Max sequence length | 128 |
| Optimizer / scheduler | AdamW / linear |
| Training infra | HF accelerate, ~90 GPU hours |
| Deployment | merged and quantized adapters |

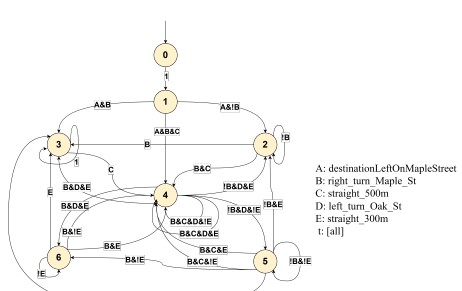

A: right_turn
B: straight_500m
C: left_turn
D: straight_1km
E: arrive_destination
t: [all]

A: destinationLeftOnMapleStreet
B: right_turn_Maple_St
C: straight_500m
D: left_turn_Oak_St
E: straight_300m
t: [all]

Figure 4: **Left:** The safety constraint. **Right:** The Desired Task in LTL generated in the first round. (Both represented in BA format for illustrating the running example)

Table 10: Prompt for NL to LTL step in LTL Generation Part

Transform the following natural language driving instruction into an LTL formula as a professional LTL expert:
{nl_instruction}

**LTL Expression Requirements:** Include multiple nested "eventually" (F), "globally" (G), "next" (X), "implies"($->$), "equivalent"($<->$), "and"(&), "or"(|), "not"(!) operators.

As a reminder, you must use the punctuations listed above as logical operators. For example, use "!" to express "not", instead of using the natural language "not" itself directly.

**AP Expression Requirements:** Words in a single AtomicProposition should be connected using underlines (_). Any other punctuation (like "%", "~", "=", "°", etc.) is not allowed to appear in the APs.

Logically represent the navigation sequence, ensuring the expression accurately reflects the instructions and conditions.

Pay attention to the matching relationships between the parentheses.

**Must obey:** Output only the raw LTL formula in one line with no explanation, no formatting, no quotes, no extra information, text display symbols, such as (*#). Sentences like "Here is the transformed LTL formula:" are not allowed. Only pure LTL formula is needed.

**One example is as following:**
**Natural language:** Go straight 1.2km through tunnel, turn left after light tower, final stop is 200m right.
**Output:** G(darkness -> headlights_on) & F(straight_1.2km -> F(lighttower & X(left_turn))) -> X(right_200m -> F(arrive))

Table 11: Prompt for AP matching

Please match the following LTL's atomic propositions: {`LTL1`}
to a pre-defined library of atomic propositions: {`atomic_proposition_library`}.
If any proposition is similar to one in the library, replace it with the library's proposition.
Only modify the atomic propositions. Do not change any logical structure or operators.
Return the updated raw LTL formula only. No explanation, no formatting, no punctuation.

Table 12: Prompt for counterexample analysis with LLM-as-a-Critic

Analyze the RABIT tool output to identify counterexamples and propose flexible changes to LTL1, ensuring the language of `input_BA` is a subset of `comparison_BA`. You need to work as an expertise professor in LTL and Büchi Automata.

**Inputs:**
- LTL1_Formula: {`LTL1`}
- Input_BA: {`input_BA`}
- Comparison_BA: {`comparison_BA`}
- RABIT_Output: {`checking_output`}
- Comparison_LTL: {`comparison_LTL`}
- Original_Natural_Language_Instruction: {`nl_instruction`} Logically represent the navigation sequence, ensuring the expression accurately reflects the instructions and conditions. Ensure that the movements of the vehicle are reflected in the LTL.

**IMPORTANT:** If natural language instruction is provided, ensure that any suggested modifications to LTL1 preserve the semantic correctness and intent of the original natural language instruction. The corrected LTL should still accurately represent the driving behavior described in the natural language.

**Steps:**
1. Analyze Counterexamples: For each counterexample, explain why it's accepted by `input_BA` but rejected by `comparison_BA`, and identify the issue in LTL1 causing the discrepancy.
2. Diagnose LTL1 Issues: Identify problematic parts of LTL1 (e.g., transitions, constraints, temporal logic) and their interaction with states and transitions in both automata.
3. Consider Natural Language Semantics: If provided, ensure proposed changes maintain semantic alignment with the original natural language instruction.
4. Propose Adjustments: Suggest minimal but flexible changes to LTL1 to resolve issues, keeping the formula simple and avoiding unnecessary complexity. Provide guidance for correction but not a concrete LTL formula.
5. Ensure Alignment: Confirm that proposed changes address all counterexamples and improve LTL1's compatibility with `comparison_BA`.

**Output:**
1. Counterexample_Analysis: "Sequence": "counterexample_sequence", "Reason": "why accepted by input_BA but rejected by comparison_BA", "Issue_in_LTL1": "issue in LTL1"
2. Proposed_Adjustments: "Adjustment": "change to LTL1", "Justification": "how it resolves the discrepancy"
3. Natural_Language_Alignment: "Semantic_Check": "ensure changes preserve original natural language intent"
4. General_Guidance: "Summary of the approach and its broader impact"

**Tips:**
1. The 't' in BA is a specially string, it's not a true alphabet string, but a signal of 'unconditional', so don't add anything about 't' to the LTL directly, instead you should think about its logic and change that.
2. Don't add operators like '$<$', '$>$' and '$=$', which are not allowed by LTL's syntactic rules.

Table 13: Prompt for LTL correction based on counterexample analysis

The current LTL1: {`LTL1`}
does not satisfy the Inclusion Check based on the Büchi Automaton comparison. Below is the analysis and revision guidance: {`understanding_output`}

Your task is to modify and simplify LTL1 to ensure that it satisfies the Inclusion Check while keeping the formula as concise and flexible as possible. Adjustments should focus on resolving the specific issues identified in the guidance rather than adding restrictive or overly complex conditions. Aim for an intuitive and streamlined solution that aligns LTL1 with the comparison Büchi Automaton.

**Instructions:**
1) Modify LTL1 based on the provided analysis.
2) Ensure the revised formula directly addresses the counterexample(s) while improving alignment with the comparison Büchi Automaton.
3) Avoid adding unnecessary constraints or increasing the formula's complexity—prioritize simplicity and precision.
4) The 't' in BA is a specially string, it't not a true alphabet string, but a signal of 'unconditional', so don't add anything about 't' to the LTL directly, instead you should think about its logic and change that.
5) Don't add operators like '<', '>' and '=', which are not allowed by LTL's syntactic rules.
6) Pay attention LTL should only contain: "goStraight", "turnLeft", "turnRight", "reachDestination"
7) Semantic Preservation: The original natural language instruction is: "{`nl_instruction`}" Logically represent the navigation sequence, ensuring the expression accurately reflects the instructions and conditions. Ensure that the movements of the vehicle are reflected in the LTL.

**Output Format:**
Must obey: Output only the raw updated LTL formula in one line with no explanation, no formatting, no quotes, no extra information, text display symbols, such as (*#).
Don't output any other things like: "LTL1", "Here is the corrected LTL1:", "The corrected LTL1 is:", "The LTL1 is:", "The LTL1 is as follows:", "The LTL1 is as follows below:" and so on.

Table 14: Analysis Provided by *LLM-as-a-Critic*

**Counterexample Analysis:**
*Sequence:* The counterexample sequence is '!straight_200m'
*Reason:* The counterexample sequence is accepted by 'input_BA' because there is a transition from state 0 to state 8 when the condition '!straight_200m' is met. However, in 'comparison_BA', there is no transition that allows the automaton to move from state 0 to any other state given the condition '!straight_200m'. The automaton in 'comparison_BA' can only stay at state 0 when the condition '!straight_200m' is met.
*Issue in LTL1:* The LTL1 formula assumes that the vehicle can go straight for 200 meters at any time, which does not align with the 'comparison_BA'.
**Proposed Adjustments:**
*Adjustment:* Remove the assumption that the vehicle can go straight for 200 meters at any time. Instead, add a condition that the vehicle can only go straight for 200 meters when it is at a specific location or state.

