# OpenReview forum: "Automatic Generation of Safety-compliant Linear Temporal Logic via Large Language Model: A Self-supervised Framework"
_ICLR.cc/2026/Conference — Submitted to ICLR 2026_

### Official Review · Reviewer_GYn2 · 2025-10-19

**Soundness:** 2
**Presentation:** 3
**Contribution:** 2
**Rating:** 4
**Confidence:** 4

**Summary:**

The paper introduces AutoSafeLTL, a self-supervised framework for automatically generating safety-compliant Linear Temporal Logic (LTL) specifications from natural language task descriptions.
Unlike prior NL2LTL systems that only translate text into logical formulas, AutoSafeLTL ensures that generated formulas formally satisfy predefined safety constraints. Using automata-based inclusion checking, the system verifies and repairs LTL outputs until they pass all safety constraints. Experiments in traffic-navigation scenarios show 0% violation rate, outperforming GPT-4 and nl2spec baselines while maintaining edge efficiency and data privacy.

**Strengths:**

- Novel integration of LLMs with formal verification:

The inclusion of automata-based inclusion checking and counterexample-guided refinement provides genuine safety guarantees missing from previous NL2LTL work.

- Practical and privacy-preserving architecture:

The cloud–edge collaboration design is well-suited for CPS applications where latency and data locality are critical.

- Strong experimental validation:

Demonstrates zero safety violations and higher efficiency than GPT-4, with comprehensive ablations across fine-tuning stages.

**Weaknesses:**

- Limited domain coverage:

The framework and datasets are restricted to navigation/traffic scenarios; generalization to other CPS domains remains unproven.

- Dependence on predefined safety constraints:

The system assumes an existing library of safety LTL rules, limiting adaptability to new or evolving domains.

- Scalability and verification overhead:

The iterative counterexample-guided repair loop may be computationally expensive for complex LTLs or multi-agent settings.

- Evaluation dataset limitations:

The repair process prioritizes safety compliance over the user’s original intent, but lacks a quantitative or qualitative measure of “semantic drift.” This could lead to safe yet semantically distorted outputs. The fine-tuning and evaluation datasets are relatively small (200–300 curated examples), synthetic, and domain-specific. Real-world instructions with linguistic ambiguity are not extensively tested.

**Questions:**

Can the framework be extended to automatically derive or learn safety rules from natural language corpora, rather than relying on manually predefined sets?

When safety repair alters the original task intent, how do you measure or control semantic drift between the initial description and the final safe LTL?

How does the system perform when handling multiple interacting constraints or large-scale multi-agent systems? Are there strategies to parallelize or approximate inclusion checking for scalability?

---

> ### Author Response · Authors · 2025-12-02
>
> Thank you for the thoughtful feedback.
>
> Q1 (Can the framework learn safety rules automatically)
>
> Yes. Our framework is compatible with automatically derived/learned safety rules, because the verification-in-the-loop pipeline is modular, extensible, and transferable: as long as safety rules can be expressed (or distilled) into formal constraints (e.g., LTL over the domain APs), they can be plugged into the same inclusion-check-and-repair loop.
> In this work we intentionally use predefined safety constraints because safety rules are high-stakes and typically require strong auditability and governance. Moreover, existing efforts on formalizing constraints are still largely limited to domains with clear, well-specified regulations (e.g., traffic), where rules can be reliably curated and reviewed. For these reasons, we do not claim automatic rule extraction as a contribution here.
> That said, we agree that learning constraints from large-scale natural-language corpora (e.g., policies, manuals, incident reports) is a promising direction to improve autonomy: a model could induce candidate safety restrictions, which can then be formalized and validated by our framework before being used for safety-compliant NL→LTL generation.
>
> Q2 (semantic drift)
>
> (1) Experimental results (semantic faithfulness).
> We have added a semantic faithfulness evaluation for all three stage-tuned models on a 1,000-instance traffic navigation NL dataset. Using the consistency metric and protocol from NL2TL (Chen et al., arXiv:2305.07766), we obtain: Stage-1: 94.7%, Stage-2: 92.5%, Stage-3: 91.1% consistency. While there is a slight decline across stages, the faithfulness remains high (>91%), indicating that our framework prioritizes safety compliance while still preserving user intent to a large extent.
>
> (2) How we measure semantic drift.
> Our consistency-based assessment checks whether the core task points in the natural-language instruction (i.e., the key objectives/waypoints of the task) are still expressed in the final LTL after safety repair. Drift is counted when these core intent elements are missing or altered in the final specification.
>
> (3) How we control semantic drift.
> •	Already implemented: our Stage-1 fine-tuning focuses on improving NL→LTL translation capability, which anchors the generated specification to the original intent and reduces drift before subsequent safety-oriented repairs.
> •	Planned extension: we can add a lightweight pre-output check that verifies the preserved core task points in the final LTL. If the check fails, we trigger another repair round (or, if no safety-compliant solution exists under current rules, we can explicitly report infeasibility rather than silently changing the intent
>
> Q3 (large-scale system)
>
> To support multiple interacting constraints at scale, we added an experiment with a large rule library: 1,000 NL tasks with 1,814 safety constraints. In such settings, verifying against all constraints is unnecessary because many rules are irrelevant to a given task. We therefore introduce a retrieval-first verification strategy: we train a lightweight BERT/ONNX retriever to select the most semantically similar safety constraints for each task (the top-K is configurable based on the desired safety level and latency budget), and we run inclusion checking only on this retrieved subset.
>
> For scalability, we also implement multi-threaded batching at two levels: (i) parallel processing across different NL tasks, and (ii) parallel verification across the multiple constraints matched to the same task. With top-5 retrieved constraints and batch size = 50, we achieve an average 0.27s per instance per iteration round for the verification step, demonstrating that the framework remains feasible under large constraint libraries when combined with retrieval-based approximation and parallelism.

---

### Official Review · Reviewer_y2dd · 2025-10-31

**Soundness:** 3
**Presentation:** 2
**Contribution:** 2
**Rating:** 2
**Confidence:** 4

**Summary:**

The paper presents AutoSafeLTL, a framework that integrates a fine-tuned LLM with cloud based agents and formal verification tools to generate LTL specifications from natural language that are guaranteed to be compliant with pre-defined safety constraints. The proposed three-stage fine-tuning strategy and the cloud-edge collaborative architecture for iterative, counterexample-guided repair are novel and interesting concepts. The authors report a 0% violation rate on their benchmark.

**Strengths:**

The core vision of tightly integrating large language models with formal verification to provide hard guarantees is timely and important. The move beyond just measuring translation accuracy to ensuring formal safety properties is a significant step forward for the application of LLMs in critical systems.

The framework strives for a fully automated pipeline from natural language to a verified LTL formula, which is an ambitious and worthwhile goal. The proposed improvement to the counterexample path extraction algorithm to enable this automation is a good contribution.

The idea of using a formal counterexample from the inclusion check not as a mere failure signal, but as a structured feedback mechanism to guide an LLM creates a meaningful dialogue between formal methods and LLMs.

**Weaknesses:**

- Lack of temporal efficiency: lack of time performance metrics. The proposed framework involves multiple, iterative loops of LLM inference, LTL-to-automata conversion, and language inclusion checking. Given that the primary application domain is safety-critical CPS (exemplified by a traffic navigation scenario), where decision-making must occur in milliseconds or seconds, the latency measurements are important. The average of 4.3 semantic correction iterations (Table 7) suggests a potentially prohibitive computational and communication overhead. Without data on execution time, it is impossible to evaluate the framework's practicality for any real system.

- Over-reliance on pre-defined, exhaustive Safety Constraints: the entire safety guarantee hinges on the assumption that the set of safety constraints Ψ is complete and correct. The paper does not address the "open-world" problem:

What happens when a user introduces a novel Atomic Proposition (AP) not present in the safety library? The LLM-as-an-Aligner might map it incorrectly or drop it, leading to semantically incorrect or unsafe specifications.

An AP generated as go_straight_200m might need to match a safety library AP like straight_200m. This is a simple case. However, more complex divergences are likely. For example, the user's instruction "Navigate to the emergency bay" might generate an AP reach_emergency_bay, while the safety constraints use a more technical term like at_service_area. The LLM may fail to link these conceptually similar but lexically different terms.

How does the system handle unforeseen scenarios or edge cases not covered by Ψ? The guarantee of 0% violations is only valid with respect to the known constraints, which provides a false sense of security if the constraint set is incomplete.

- Computational scalability: the paper ignores the well-known state explosion problem in automata-based verification. The complexity of BA construction and inclusion checking is exponential in the number of APs. The evaluation uses a small, manageable set of APs (4-8). The framework's behavior and feasibility with a realistic number of APs (e.g., 30+ for a real autonomous driving system), are not discussed. This raises serious doubts about its scalability.

- Benchmark: the evaluation is conducted on a custom benchmark. There is no demonstration of generalization, raising concerns about overfitting to the authors' specific data style (Chen et al., 2023).

- Missing baselines: the paper fails to compare against established NL2LTL methods, such as grammar-based approaches (e.g., Fuggitti & Chakraborti, 2023), other LLM-based fine-tuning methods (e.g., Chen et al., 2023). Consequently, it is impossible to determine if the significant complexity of AutoSafeLTL provides any tangible benefit over existing, potentially simpler and more efficient methods.

- Data privacy: the paper emphasizes "data locality for privacy" as a key advantage of the edge-side User LLM. However, the generated LTL formula, which is a direct formalization of the user's intent and the environmental context, is sent to the cloud for the verification and alignment steps. While the raw natural language may stay on the device, the formalized semantic content does not. This nuanced privacy trade-off is not adequately discussed.

Fuggitti, F., & Chakraborti, T. (2023). NL2LTL – A Python Package for Converting Natural Language (NL) Instructions to Linear Temporal Logic (LTL) Formulas.

Chen, Y., Gandhi, R., Zhang, Y., & Fan, C. (2023). NL2TL: Transforming Natural Languages to Temporal Logics using Large Language Models.

**Questions:**

What is the end-to-end latency of your framework, from receiving the NL input to outputting a safety-compliant LTL, on your benchmark? Please provide end to end average and broken down by component average.

How does your system behave if a user input contains an AP not in the pre-defined library? Please provide a concrete example and explain the process. What happens in the case of an AP that is not just new, but semantically incompatible with the safety rules encoded in the constraints? Does the framework simply suppress the action, or is there a way to inform the user that their request is inherently unsafe given the current rules? Please walk through the specific system behavior in such a scenario.
What is the maximum number of APs your framework can handle while maintaining reasonable verification times? Can you show how verification time grows with the number of APs?

Why did you not compare your method against other specialized NL2LTL tools? (e.g. Chen et al., 2023,  Fuggitti & Chakraborti, 2023) Can you provide such a comparison on a publicly available NL2LTL benchmark to demonstrate superior accuracy or efficiency? (e.g. Chen et al., 2023)

Fuggitti, F., & Chakraborti, T. (2023). NL2LTL – A Python Package for Converting Natural Language (NL) Instructions to Linear Temporal Logic (LTL) Formulas.

Chen, Y., Gandhi, R., Zhang, Y., & Fan, C. (2023). NL2TL: Transforming Natural Languages to Temporal Logics using Large Language Models.

---

> ### Author Response · Authors · 2025-12-02
>
> Thank you for the thoughtful feedback.
>
> Q1 (End-to-end latency) We evaluate latency on a 1,000-instance NL benchmark to reflect large-scale, near-real-time verification (not a toy set). With batch size = 20 and 5 safety constraints verified in parallel, our system averages 0.27s per instance per iteration. Given 4.3 average rounds to reach a safety-compliant LTL, this is ~1.16s end-to-end per NL input on average.
> Per-round component profiling:
>
> 1) User LLM generation/repair (NL→LTL): 0.760 s
> 2) BA construction (Spot, parallel): 0.0097 s
> 3) Safety verification (RABIT inclusion checks, parallel): 0.0227 s
> 4) Other overhead (parsing/AP mapping/bookkeeping): ~0–0.185 s (pipeline-dependent)
>
> Overall, these results support the practicality of verification-in-the-loop NL→LTL generation at scale.
>
> Q2 (AP related)
>
> (1) If the user input contains an AP not in the pre-defined library.
> Our AP-matching prompt is intentionally strict: the model is instructed to only replace a token with an AP from the library when it is highly similar; otherwise, it keeps the original token unchanged. This prevents “hallucinated” remapping. At the automata level, AP naming itself is not critical, but it is crucial for counterexample-guided regeneration: readable, semantically grounded APs help the LLM interpret counterexample evidence and propose correct repairs.
> Concrete example. Suppose the library contains turn_left, turn_right, keep_lane, stop, but the user writes “make a U-turn at the next junction,” producing a token u_turn not in the library. Since there is no high-similarity library AP, the system keeps u_turn unchanged.
>
> (2) New but semantically incompatible AP w.r.t. safety rules.
> Pure inclusion checking cannot detect “semantic incompatibility” between symbols (e.g., a token that conceptually contradicts safety intent) because automata-level verification reasons over system behaviors driven by symbols, not the external meaning of the symbols themselves. Therefore, we handle AP-level incompatibility explicitly via the AP library: we can attach a lightweight lexicon/policy layer over AP strings (e.g., forbidden actions/unsafe intent keywords) to flag contradictions before or during repair. In this case, rather than silently suppressing the request, the framework can surface an explicit message that the request is inherently unsafe under the current rules and cannot be satisfied without relaxing constraints (i.e., “no safety-compliant specification exists”). Operationally, this is triggered when (i) the AP is flagged by the AP-policy lexicon or (ii) repeated repair reaches the maximum rounds with persistent violation, at which point we report unsatisfiability under the current constraint set.
>
> (3) Maximum AP scale and AP explosion concerns; growth of verification time.
> Our current benchmark formulas use 3–15 unique APs (median 9), with length 55–267 characters (median 167), nesting depth up to 8, and 2–10 temporal operators (median 6), showing our system already handles at least 15 APs in practice with near-real-time verification.
> We acknowledge that BA construction and inclusion checking have worst-case exponential dependence on the number of APs. This is precisely why we adopt RABIT for inclusion checking: it incorporates automata optimizations (e.g., AP pruning/adding–style reductions) that mitigate AP blow-up in practice.
>
> Q3 (Comparison)
>
> We did consider these specialized NL2TL/NL2Spec lines of work and have already leveraged them in our pipeline design. In particular, we repeatedly cite Chen et al. and follow their data construction methodology to build our fine-tuning dataset; this is reflected in Table 5 as “LLaMA-35k-lifted”, which is derived by lifting their generation scheme into our setting.
>
> However, a direct head-to-head comparison with Chen et al. is not methodologically clean for our target: their work focuses on generating STL (Signal Temporal Logic) specifications, whereas our framework generates LTL specifications. Because STL and LTL differ in semantics and evaluation protocols (e.g., quantitative robustness for STL vs. ω-regular language semantics for LTL), comparing “accuracy” across these logics would require additional translation/normalization assumptions that may confound conclusions. For this reason, we adopted their ideas for dataset construction and evaluation design, but did not claim a direct baseline comparison.
>
> More broadly, our paper’s primary objective is early-stage safety-compliant specification generation (i.e., generating a task LTL that is provably compliant with a set of safety constraints via formal verification), rather than optimizing unconstrained semantic matching alone. Many NL2LTL tools are evaluated mainly on semantic fidelity to a reference formula, without the additional safety-compliance requirement that defines our setting. We do agree that a comparison on a public NL2LTL benchmark would be valuable.

---

### Official Review · Reviewer_Uwiw · 2025-11-01

**Soundness:** 2
**Presentation:** 2
**Contribution:** 1
**Rating:** 2
**Confidence:** 5

**Summary:**

This work proposed a framework to generate LTL based on natrual language description with saferty complience.

**Strengths:**

This work is focusing on an important problem in CPS research, and it is critical in safety-critical applications.

**Weaknesses:**

My major concern is the originality of this work and the novelty.

It is not novel that connects a external verification module to the contents generated by LLM, I think this work is just another instance of these works. The difference is the out of LLM is LTL itself in this work,and others' output should follow some formal specifications such as LTL and first-order logic. It is very intuitive to use the checking results as feedback or guidance for LLM, and this is not novel.
Some early attemts can be found below:
1. Counterexample Guided Inductive Synthesis Using Large Language Models and Satisfiability Solving (MILCOM'23)
2. Assuring LLM-Enabled Cyber-Physical Systems (ICCPS'24) (This work has open-sourced their implementation as well (https://weizhesyr.github.io/SafePilot_doc/))

My second concern is the evaluation, it is easier for conjuction formulas but harder for disjuction in general. And it is easier for global operator but harder for until operator, I think it is more convicing if these statistics can be shown and how complex LTL this framework can handle. Currently, these details are missing, the statistics in Table 8 is insufficient.

There are some minor issues for clarity as well. For example, it is unclear why LLM-as-a-critic is useful, the only statement is they discoverd this and this not validated in ablation study. Since it requires multiple rounds for a correct LTL generation, it is unclear why reasoing model is not tested.

**Questions:**

N/A

---

> ### Author Response · Authors · 2025-12-02
>
> Thank you for the thoughtful feedback.
>
> Q1 (Novelty / Originality) We agree that “LLM + external verifier + iterative feedback” is a broad paradigm. However, our paper is not positioned as another instance of attaching a verifier to LLM outputs.
> Goal difference (what problem we solve). In CPS, formal specifications (e.g., LTL) are a primary interface for downstream synthesis/verification, but non-expert users cannot realistically author correct LTL. Existing NL2LTL work largely emphasizes semantic alignment (i.e., matching the intended task), while not enforcing external/environmental safety constraints during specification authoring. We target early-stage safety guarantees at the specification level: the generated task specification must be provably safety-compliant w.r.t. predefined safety rules (rather than checking safety only after the specification is used later in planning/testing).
> LLM-guided regeneration is simply a practical mechanism to leverage LLMs’ generation/understanding ability inside this safety-first specification process, not the claimed novelty.
> How this differs from the two cited systems.
>
> MILCOM’23 (CEGIS + SMT for planning/program synthesis). That work uses LLMs as a learner inside a counterexample-guided loop to synthesize plans/programs verified by an SMT oracle.
> 1) Output artifact: plans/programs (not a safety-compliant specification intended for reuse).
> 2) Verification form: SMT-based counterexamples for candidate plans vs given specifications.
> 3) Our difference: we generate/repair the LTL specification itself and guarantee safety via automata-theoretic language inclusion; furthermore, we contribute counterexample-path extraction to make inclusion failures actionable for automated LTL repair.
>
> CCPS’24 (Assuring LLM-enabled CPS via verified planning). This framework positions the LLM as a planner/decision-maker and uses Z3/SPOT to verify the generated plan against safety/temporal constraints, then re-prompts using invalid steps.
> 1) Primary objective: assure LLM-generated plans for execution.
> 2) Treatment of specifications: it prompts the LLM to produce logic formulas for requirements, but explicitly notes converting LTL to automata is done after expert review.
> 3) Our difference: we do not aim to make the LLM a planner. We aim to translate a user’s given task in natural language into a non-violating (safety-compliant) LTL specification in a fully automated way, so the specification itself is safe and can be reliably used by downstream CPS toolchains.
>
> Q2 (Evaluation) We agree that LTL difficulty varies by Boolean structure (∧ vs ∨) and temporal operators (e.g., G vs U). To address this, we analyzed our 300-instance benchmark and will expand Table 8 to report operator/complexity statistics. On this dataset, conjunction appears in 298/300 (99.3%) formulas, while disjunction appears in 111/300 (37%) (and always co-occurs with conjunction). For temporal operators, G appears in 292/300 (97.3%), whereas U appears in 118/300 (39.4%). In terms of complexity, formulas range from 55–267 characters (median 167), use 3–15 unique APs (median 9), and have a maximum parenthesis nesting depth of 8; the number of temporal operators per formula ranges from 2–10 (median 6).
>
> Q3 (LLM-as-a-critic) Regarding the comment that the Critic is not validated via ablation, we have in fact included this study: in Table 7, AutoSafeLTL-D corresponds to the LLM-as-a-critic ablation (i.e., removing the Critic). The results show that the full AutoSafeLTL requires 4.3 interaction rounds on average, while removing the Critic increases this to 11.0 (with 0% violations in both cases). This demonstrates that the Critic significantly reduces the number of iterations and improves repair efficiency, by converting verification-failure evidence into more actionable repair instructions.

---

### Official Review · Reviewer_PXvT · 2025-11-04

**Soundness:** 2
**Presentation:** 2
**Contribution:** 2
**Rating:** 4
**Confidence:** 4

**Summary:**

This paper proposes AutoSafeLTL, a self-supervised framework for converting natural language task descriptions into safety-compliant Linear Temporal Logic (LTL) formulas. Unlike previous NL2LTL approaches that focus on translation accuracy, AutoSafeLTL integrates formal verification into the generation loop to guarantee that outputs comply with predefined safety constraints.

**Strengths:**

1. Integrating formal verification directly into the language model generation loop is interesting.
2. Ablation studies and comparisons against GPT-4 and NL2Spec demonstrate measurable benefits, which have lower iteration counts and zero violation rates.

**Weaknesses:**

1. While the multi-stage fine-tuning strategy is practical, it mainly repurposes standard LLaMA fine-tuning with LoRA adapters, which is not a significant methodological innovation.
2. Will the approach generalize to other CPS domains (e.g., robotics?).
3. There is no assessment of whether generated LTL formulas remain semantically faithful to user intent after repair. The authors used safety compliance as the metric to measure, but alignment quality is not measured.

**Questions:**

Please clarify the concerns raised in the weaknesses

---

> ### Author Response · Authors · 2025-12-02
>
> Thank you for the thoughtful feedback. Below we clarify the concerns in the weaknesses. Firstly, We emphasize that our contribution is not a new parameter-efficient fine-tuning method; LoRA/QLoRA is simply a deployment-oriented choice. The novelty lies in making formal verification (inclusion counterexamples and extracted paths) usable for fully automated LTL generation under a cloud–edge alignment loop.
> W1 (LoRA not innovation). We agree that LoRA/QLoRA is not our contribution; it is only an efficiency choice for the edge “User LLM” implementation. Our contributions are: (i) formal-verification-in-the-loop generation/repair with language-inclusion safety checking, (ii) counterexample-path extraction to make verification failures actionable for automated repair, and (iii) a cloud–edge Aligner/Critic loop that turns formal evidence into repair guidance. Regarding the three-stage training, we do not claim a new fine-tuning algorithm; rather, the novelty is the problem decomposition + verification-guided curriculum (S1 translation, S2 syntax repair, S3 semantic/safety repair) that makes a compact edge model effective for this formal task.
>
> W2 (Generalization). The method is domain-agnostic: given (a) domain APs and (b) safety constraints in LTL, the same pipeline (LTL→BA, inclusion check, counterexample path, repair) applies unchanged. Traffic is just an instantiation for evaluation; robotics/CPS can be supported by redefining APs/constraints.
>
> W3 (Semantic faithfulness / intent preservation). We have added a semantic faithfulness evaluation for all three stage-tuned models on a 1,000-instance traffic navigation NL dataset. Using the consistency metric and protocol from NL2TL (Chen et al., arXiv:2305.07766), we obtain: Stage-1: 94.7%, Stage-2: 92.5%, Stage-3: 91.1% consistency. The slight decrease across stages matches our design goal: later stages increasingly prioritize safety-compliant repair when conflicts arise. Importantly, even with this safety-first objective, the models retain high semantic faithfulness (>91%), indicating limited drift from user intent in practice.

---

### Meta-Review · Area_Chair_Rs7y · 2026-01-07

**Summary:**

The paper presents AutoSafeLTL, which turns natural language into LTL, then uses automata inclusion checking to iteratively “repair” the formula until it satisfies a safety-rule library. Reviewers like the safety-first framing and the strong zero-violation results. However, the main concerns are limited novelty over prior LLM+verifier repair loops, a narrow custom traffic benchmark with missing NL2LTL baselines, and unclear scalability/generalization. The rebuttal adds latency and intent-faithfulness, but key doubts remain. Recommend rejection.

**Reviewer Concerns:**

- Rebuttal addressed: (1) concerns about intent drift and practicality by adding semantic-faithfulness results, (2) end-to-end latency profiling, (3) an ablation showing the Critic reduces iterations.

- Still outstanding: (1) novelty vs prior LLM+verifier repair work, (2) missing comparisons to established NL2LTL tools/benchmarks, (3) limited evidence for generalization/open-world AP handling.

**Reviewer Scores:**

- PXvT: likely would have raised to a weak accept since the rebuttal directly addressed LoRA “novelty,” added a semantic-faithfulness evaluation, and clarified the intended contribution.
- Uwiw: likely would have kept a reject because their main objection is lack of novelty vs prior LLM+verifier loops, and they expressed very high confidence.
- y2dd: likely would have kept score or raised slightly due to the new latency breakdown, but still below threshold given missing baselines, open-world constraint/AP concerns, and scalability questions.
- GYn2: likely keep the score since the rebuttal provides scalability/latency discussion, but only partially easing their main concerns.

---

### Decision · Program_Chairs · 2026-01-26

Reject